# Do People Want the ‘New Normal’? A Mixed Method Investigation of Young Person, Parent, and Clinician Experience and Preferences for Eating Disorder Treatment Delivery in the Post-COVID-19 World

**DOI:** 10.3390/nu15173732

**Published:** 2023-08-25

**Authors:** Julian Baudinet, Anna Konstantellou, Ashlea Hambleton, Katrin Bialluch, Georgina Hurford, Catherine S. Stewart

**Affiliations:** 1Institute of Psychiatry, Psychology and Neuroscience (IoPPN), 16 De Crespigny Park, London SE5 8AD, UK; 2Maudsley Centre for Child and Adolescent Eating Disorders (MCCAED), Maudsley Hospital, De Crespigny Park, London SE5 8AZ, UK; 3InsideOut Institute, Central Clinical School, Faculty of Medicine and Health, Charles Perkins Centre (D17), University of Sydney, Camperdown, NSW 2006, Australia

**Keywords:** adolescent, eating disorder, anorexia nervosa, bulimia nervosa, family therapy, COVID-19, virtual therapy, online therapy

## Abstract

Eating disorder treatment was predominantly provided online during the COVID-19 pandemic, which has continued into the post-pandemic world. This mixed method study explored young person, parent/caregiver, and clinician experiences of child and adolescent eating disorder treatment. In total, 90 participants (25 young people, 49 parents/caregivers, and 16 clinicians) completed online surveys about the experience of online working. Data were compared to similar data collected by the same service earlier in the pandemic. The results show that preferences are largely unchanged since 2020; online treatment is considered helpful and acceptable by all groups. Nevertheless, face-to-face assessment sessions (young people: 52.2%; and parents/caregivers: 68.9%) and final sessions (young people: 82.6%; and parents/caregivers: 82.2%) were preferred compared to online. There was also a preference for early treatment sessions to either be always or mostly face-to-face (young people: 65.2%; and parents/caregivers: 73.3%). The middle and latter parts of treatment were a time when preferences shifted slightly to a more hybrid mode of delivery. Participants reported finding engagement with the therapist (young people: 70.6%; and parents/caregivers: 52.5%) easier during face-to-face treatment. Stepping away from the binary of online *or* face-to-face, the current data suggest that a hybrid and flexible model is a way forward with current findings providing insights into how to structure this.

## 1. Background

The novel corona virus (COVID-19) pandemic required eating disorder services internationally to quickly pivot in early 2020 from predominantly face-to-face treatment delivery to a largely online provision of services. Pre-pandemic studies evaluating the online implementation of evidence-based mental health treatment was in its relative infancy prior to this. The available data consistently demonstrated comparable efficacy between online and face-to-face delivery for a range of mental health difficulties [1,2], including eating disorder treatment [3,4,5]. Yet, the rapid transition to treatment via videoconferencing platforms, termed telehealth, during COVID-19 lockdowns exposed not only the strengths but also the challenges and downfalls of virtual treatment delivery. Patients and clinicians were thrust into this change with little preparation, relying on a limited amount of research, expert guidance, communication with other mental health professionals, and technical competence [6,7,8,9,10].

Overall, research suggests that clinicians were able to successfully adapt the treatment they were providing, ensuring the continuity of a high standard of care with good treatment outcomes [11]. In a service evaluation of a specialist child and adolescent eating disorder service, young people, as well as their parents and carers, pinpointed unique benefits of online treatment during the pandemic, such as improved e-communication and increased comfort. Participants reported that receiving treatment in the home reduced pre-appointment apprehension and helped young people feel more at ease discussing sensitive topics with clinicians. Parents further noted socioeconomic benefits, including reduced travel costs, and clinicians reflected on how virtual platforms improved ease of access [12]. Longer pre-lockdown treatment duration, as well as a stronger therapeutic relationship, were associated with a more positive perspective of the shift towards online therapy [13]. Downfalls of online therapy noted by both eating disorder professionals, as well as patients and their families, included altered relational experiences and an increased pressure for families to monitor physical health risk in the home [12]. 

Prior to the COVID-19 pandemic, research has demonstrated that participants preferred face-to-face therapy compared with ‘e-therapy’ alternatives, although many expressed motivations to try e-therapy in the future [14]. Despite the successful adaptation of online eating disorder treatment for young people during the pandemic [12,15,16,17], an audit of a specialist child and adolescent service in North London suggests these preferences remain largely unchanged [12]. Similarly, a pilot examination of 63 adult patients with eating disorders in Israel, whose treatment moved online in 2020, found that most patients viewed virtual treatment as a ‘situation specific necessity’ and would not choose to continue with this form of treatment [13]. In this study, age was not correlated with this perspective, suggesting that young people may have similar views. Other available data on patient preferences suggest young people prefer face-to-face treatment delivery more so than their parents/caregivers [15,16]. 

Several years post-lockdowns, it appears that COVID-19 has sparked a permanent transformation in treatment delivery in the UK and internationally. Online or hybrid treatment delivery is more commonplace and experts have recommended that flexible treatment delivery, such as the use of online platforms, will be key to increasing early intervention efforts [18]. This seems to match a broader cultural shift internationally for more of our daily interactions to be online. There have been huge increases in the use of online platforms for social, education, vocational, and healthcare interactions. While beneficial and more convenient, many have reported a sense of fatigue, exhaustion, and disconnection as a result of being online more [19,20]. As such, it is important to understand people’s experiences of and preferences for being online.

Within child and adolescent eating disorder services specifically, many UK-based National Health Service (NHS) teams continue to offer online treatment to patients despite COVID-19 restrictions easing. Given this, there is a need to better understand patient, parent/caregiver, and clinician preferences further, especially now that the receipt of online therapy is more determined by choice, rather than enforced by health restrictions. Individual patient and clinician views need to be considered to ensure the care provided is acceptable, effective, and patient centred [13]. Negative expectations, attitudes, and existing preferences of treatment can all adversely affect the implementation of online therapy [14] and should, therefore, be considered in a culture that strongly advocates for the continued use of telehealth beyond the pandemic [21]. The authors are not aware of any published studies reporting the experience of online working in child and adolescent services post the easing of COVID-19 restrictions.

The current study sought to address this gap by eliciting young person, parent/caregiver, and clinician experiences of online therapy in an established London-based specialist child and adolescent eating disorder service that is offering both online and face-to-face treatment appointments. It is a repeat of an earlier study in the same service [16], which examined experiences during the early parts of the COVID-19 pandemic, when online treatment was enforced due to risk of COVID-19 infection. The primary aim of this study was to assess young person, parent/caregiver, and clinician current experiences of online specialist child and adolescent eating disorder treatment now that COVID-19 restrictions on face-to-face meetings have eased. The secondary aim was to compare this to similar data previously collected by the same service in the early parts of the COVID-19 pandemic.

## 2. Materials and Methods

### 2.1. Study Design

This cross-sectional study employed an online survey methodology to anonymously gather information about the experience of and preferences for online and face-to-face treatment. The use of online surveys to collect both quantitative and qualitative data was considered most appropriate given anonymity of participants was preserved and disclosure of ones’ own preferences encouraged. It also allowed for a more flexibility and convenient approach to data collection [22].

### 2.2. Sample

A convenience sample of all young people and parents open for treatment at the Maudsley Centre for Child and Adolescent Eating Disorders (MCCAED) outpatient service in July 2022 was used in this study. Inclusion criteria were (a) being a young person with a diagnosis of a DSM-5 [23] eating disorder or their parent/caregiver, (b) a current patient of the outpatient service at MCCAED, and (c) having been in treatment for a minimum of two months. Having young people in treatment for at least two months allowed enough exposure to online and/or face-to-face treatment for responses to be meaningful. All young people were less than 18 years of age.

All clinicians who were (a) employed at MCCAED at the time of data collection and (b) were a registered mental health professional were eligible for this study and invited to participate. Clinicians were all fully qualified, registered mental health professionals, including clinical psychologists, family therapists, medical doctors, and psychiatrists, all of whom had experience in the treatment of eating disorders. Level of experience was not included as an eligibility criterion to participate, rather, these data were collected and reported on.

### 2.3. Treatment Setting

MCCAED is a specialist child and adolescent eating disorder service in South London, UK. It has a catchment area of approximately two million people and sees people up to the age of 18 years. It offers a range of services, including outpatient [24,25] and day patient [26,27] services. The primary treatment is eating disorder focussed family therapy [28,29], although a range of other treatments are also offered, including multi-family therapy [30,31] and radically open dialectical behaviour therapy [32,33], depending on individual and family need.

### 2.4. Ethical Approval and Consent

This study was approved by the South London and Maudsley (SLaM) child and adolescent mental health services (CAMHS) Service Evaluation and Audit Committee (Project #: 2022-02). As this study constitutes clinical audit or service evaluation, NHS Research Ethics Committee approval was not required. SLaM CAMHS service evaluation and audit approval allows for analysis and publication of anonymised data collected as part of approved projects without written consent from participants or caregivers. Nevertheless, all participants were informed of the design and purpose of the study and provided opportunity to speak to the project team with any queries. No young person was invited to participate without parental/caregiver awareness of the study. It was made explicit to all that participation was voluntary. All methods were performed in accordance with the stipulated guidelines and regulations.

### 2.5. Procedure

Links to online surveys for young person, parents/caregivers, and clinicians were prepared via the Qualtrics XM online platform [34]. Similar surveys were prepared for clinicians, young people, and parent/caregivers. Both Likert-scale and free text questions were utilised to assess experience of online therapy and how that compared to face-to-face. Survey questions also assessed what factors influenced their preferences. The available email addresses on file for eligible families were used for recruitment purposes. This was often a single email address per family for a parent/caregiver. As such, the email Invitation requested the survey be sent to their young person and/or partner (where applicable) if they met the inclusion criteria and they deemed it appropriate. Initial email invitations were sent in July 2022 followed by a maximum of three follow-up emails. Recruitment to this study closed in November 2022. Questions in the surveys from the original study [16] were repeated in the current surveys as well as some additional questions. This allowed for direct comparisons to be made between responses from participants at the start of the pandemic and at the end.

### 2.6. Analysis Plan

Quantitative data (Likert and categorical scale rating questions) were analysed using a series of non-parametric tests. Fischer’s exact test was used to examine associations between participant type (young person/parent-caregiver/clinician) and level of performance on various treatment components, as well as associations between participant type (young person/parent-caregiver/clinician) and preferences of treatment delivery mode in general and at different stages of treatment. Additionally, Mann–Whitney U tests were used to compare the experiences of participants in the current study, with data collected at the same service at the beginning of the COVID-19 pandemic in 2020.

Qualitative data were analysed using a reflexive thematic analysis method [35]. The data were approached from a critical realist framework, which viewed meaning and experience as subjective and influenced by social and cultural contexts. Comments were first coded, and then topics were defined. Thereafter, themes were developed through reflexive engagement with the data. Young person, parent/caregiver, and clinician data were initially analysed separately. Given the similarities in findings, data were then combined to generate themes.

Quantitative data were analysed using Statistical Package for the Social Sciences (SPSS) version 29 [36]. No software was used during the reflexive thematic analysis. Figures were created using Microsoft Excel version 16.76 [37].

### 2.7. Reflexivity Statement

The analysing authors (AK and AH) had quite different experience of child and adolescent eating disorder treatment, MCCAED as a service, and the UK healthcare system. AK (cisgendered female, Greek–British, Assistant Psychologist, and PhD) had worked in MCCAED for many years and had extensive exposure to the treatments MCCAED offers, although did not deliver any clinically. AH (cisgendered female, white Australian, Clinical Psychologist, and MClinPsy) was a visiting researcher at the time of data analysis. She approached the data with several years’ experience, offering specialist clinical treatments and conducting eating disorder research in private and public health settings in Australia. These differences promoted reflection around the impact of the social and cultural context of the team, the work, and the NHS system on treatment experiences. It also facilitated interesting discussion around the different experiences of the COVID-19 pandemic and the impact on individuals, families, and healthcare generally.

## 3. Results

### 3.1. Sample

A total of 154 families and 23 clinicians were approached to participate in the study. Responses from 90 individuals are reported below: 25 young people, 49 parents/caregivers, and 16 clinicians. Due to the anonymous nature of the survey, it was not possible to determine whether young people and parents/caregivers who completed surveys were from the same or different families.

The majority of the young people had a diagnosis of anorexia nervosa/atypical anorexia nervosa (n = 16/25, 65.00%). The remaining had a diagnosis of bulimia nervosa/atypical bulimia nervosa (n = 3/25, 12.00%), binge eating disorder (n = 1/25, 4.00%), and other specified/unspecified feeding and or eating disorder (n = 5/25, 20.00%). No further individual or family demographics were collected.

The average years of experience since qualifying for clinicians was 11.86 years (range 1–34). Half the clinicians (n = 8, 50.00%) had had some experience of providing treatment online prior to the COVID-19 pandemic, with the other half only learning to use online technology during the pandemic. Of those who did have some prior experience, most said it was very minimal (a few sessions or less). Only two clinicians reported substantial experience of online working prior to the pandemic (>1 year experience).

### 3.2. Treatment Characteristics

Most young people (n = 18/25, 72.00%) and parents/caregivers (n = 39/49, 79.59%) reported attending hybrid treatment whilst at MCCAED, a combination of some face-to-face and some online sessions (see Table 1). All parents/caregivers had attended at least one session of treatment, with most (n = 41/49, 83.7%) having attended six or more (number of sessions attended: 1–5 = 8/49, 16.33%; 6–10 = 11/49, 22.45%; 11–15 = 4/49, 8.16%; 16–20 = 7/49, 14.29%; and 20+ = 19/49, 38.78%). Aside from one young person who had only attended an assessment session, the majority had received more than 10 sessions of treatment (number of sessions attended: 1–5 = 5/25, 20.00%; 6–10 = 5/25, 20.00%; 11–15 = 3/25, 12.00%; 16–20 = 5/25, 20.00%; and 20+ = 7/25, 28.00%).

### 3.3. Quantitative Findings

#### 3.3.1. Young Person and Parent/Caregiver Experience of Online Treatment

Both young people and parents/caregivers rated their experience of online treatment as relatively high. Median scores for the overall experience (the subjective general impression of online treatment), the ability to be understood by the therapist and address important issue, and the overall benefit were all rated between 5–7/7. The impact of technology was relatively minimal for young people (2/7) and slightly higher for parents/caregivers (3.5/7) (see Table 2).

Mann–Whitney U Tests examining differences in young person and parent/caregiver experiences of online treatment revealed no significant differences on all domains except one. Parents felt more understood by the therapist compared to young people (*p* = 0.002), although both rated this quite highly, suggesting minimal impact of online treatment on this domain.

#### 3.3.2. Treatment Mode Preferences

Despite participants rating online therapy relatively highly, there was still a general preference for treatment to be delivered face-to-face. In total, 60.90% of young people and 60.00% of parents/caregivers reported an overall preference for treatment to be provided either ‘mostly face-to-face’ or ‘100% face-to-face’ (see Table 3). No parents/caregivers and only one (4.30%) young person preferred treatment to be entirely provided online. The results of the Fischer’s exact test did not indicate a significant association between participant type (young person/parent-caregiver) and mode of overall treatment preference (*p* = 0.60).

There was a strong preference for face-to-face treatment for assessment and ending sessions. However, seven (30.4%) young people did not have a preference either way regarding the assessment session (see Figure 1).

Young people generally preferred to have face-to-face treatment at all stages of treatment, whereas parent/caregiver preferences shifted more towards a combination of online and face-to-face treatment delivery at the middle and later parts of treatment (see Figure 2).

Fischer’s exact test did not indicate a significant association between participant type and mode of treatment preference at assessment (*p* = 0.36), early stages of treatment (*p* = 0.45), middle of treatment (*p* = 0.20), or final session (*p* = 0.16). Fischer’s exact test did indicate a significant association between participant type and mode of treatment preferences at later stages of treatment (*p* = 0.02).

#### 3.3.3. Perceived Impact of Treatment Mode on Different Aspects of Treatment

Both young people and parents/caregivers considered it easier to engage with the therapist and discuss emotional difficulties face-to-face compared to online. However, treatment mode did not seem to impact perceived confidentiality, the ability to discuss practical tasks, the speed of recovery, or the ability to involve family members (see Table 4).

The results of the Fischer’s exact test did not indicate a significant association between participant type (young person/parent-caregiver) and level of perceived performance on examined treatment domains, including engagement with therapist (*p* = 0.43), confidentiality (*p* = 0.36), ability to discuss emotional difficulties (*p* = 0.49), ability to discuss practical difficulties (*p* = 0.11) speed of recovery (*p* = 0.73), and involvement of family (*p* = 0.86).

#### 3.3.4. Comparison of Current Experiences of Online Therapy to Those at the Beginning of the COVID-19 Pandemic

Mann–Whitney U Tests were carried out to compare clinicians, young people, and parents/caregivers’ experience of online sessions at the beginning of the pandemic (2020 data reported by Stewart et al. [16]) and two years later (the 2022 cohort) (see Table 5). Results indicated a significant increase in level of comfort for clinicians from the start of the pandemic to two years after (*U* = 18.00, *p* < 0.001). For parents/caregivers, online therapy was rated as significantly more able to address important issues for the young person (*U* = 1160.50, *p* = 0.01), and there was a significant reduction in the impact of technology on the experience of online sessions (*U* = 420.50, *p* < 0.001). No significant differences were detected in young people between the two cohorts on any domain.

### 3.4. Qualitative Findings

Three main themes and nine subthemes were developed from the reflexive analysis. Each are described below with associated quotes transcribed verbatim.

#### 3.4.1. Theme 1: Something Gained

Clinicians, parents/caregivers, and young people discussed several gains of having therapy online, which fell into three subthemes: care continuity and increased access (1.1), convenience and comfort (1.2), and new opportunities (1.3).

##### Care Continuity and Increased Access

A perceived notable benefit of online therapy was the availability and/or continuity of care when otherwise face-to-face sessions would not be possible. Both young people and parents/caregivers were grateful of online sessions, which meant they could ‘*access therapy*’ when otherwise it would have to be missed or paused. Parents/caregivers also noted that online therapy allowed for treatment to continue when on holidays abroad or when someone was ‘*sick or therapist is working from home*’. Online therapy also meant more families had access to treatment who may not have been able to.


*‘We had to travel over the summer holidays so it was fantastic to be able to continue therapy’*
(parent/caregiver)


*‘… online [meant we had] continuity of therapy when not able to attend clinic’*
(young person)


*‘I have found that for some families and young people, especially those with neuro diverse conditions or who live far and are less able to attend in person, conducting online therapy has been more beneficial or enabled the process better.’*
(clinician)

##### Convenience and Comfort

The practical benefits of accessing treatment online without the need to travel, miss school, or disrupt family life were all considered unique advantages of online treatment. Having sessions online allowed for family life to continue with the least disruption to young peoples’ school attendance and parents/caregivers’ work commitments. Young people noted that online therapy meant they were ‘*less anxious about missing school*’, ‘*less school time missed*’, and ‘*less travelling time*’, which resulted in increased appointment frequency and being ‘*able to have them [sessions] more regularly*’. Likewise, parents/caregivers noted the reduction in ‘*time-consuming*’ travel time, which impacted their preference for online sessions.

Face-to-face sessions were the preferred overall method of treatment delivery but only if it did not disrupt family life and school attendance too significantly. School attendance was described as particularly important for older children who had exams. Having sessions online, particularly at the later stages of treatment, allowed young people to focus on rebuilding their life outside of their illness. Additionally, some young people commented how the comfort of their home fostered calmness, which led to increased engagement as some found it easier to express themselves.


*‘Face-to-face is preferred, however, sometimes online is good to discuss simple practical issues that does not require a meeting’*
(parent/caregiver)


*‘Also, there are times when travel is difficult and online is a good compromise.’*
(parent/caregiver)


*‘My own home…my dogs are there…having them sit with me whilst I have the meeting makes me feel calmer’*
(young person)


*‘Flexibility for staff and patients. Some patients more comfortable in own home’*
(clinician)


*‘[online is] more comfortable … easier to tell her [therapist] how I feel’*
(young person)


*‘Towards the end online can work better in terms of returning to normal life and not missing school’*
(clinician)

Lastly, one parent/caregiver noted how having online sessions available as an option was helpful as it meant they were ‘… *able to have meetings at short notice during crises, when travel has not always been possible’ (parent/caregiver).*

##### New Opportunities

Many participants commented on the way online sessions allowed for learning and discoveries to arise. Clinicians discussed learning new skills to adapt to delivering therapy online as well as an increase in their confidence and enjoyment in this mode of treatment delivery. One clinician noted that they had ‘*developed a real pleasure in working on-line and would not want to lose that format for therapy entirely’ (clinician).* Many clinicians described a shift in confidence and opinions regarding the effectiveness of online therapy. Another clinician noted *‘I feel more confident and more effective as a therapist working online now. There is no doubt in that’ (clinician)*. Some described how this was hard to predict this change in thinking and more something they learnt by doing: ‘*You have to experience it to believe it’ (clinician).* Clinicians also discussed a shift in belief regarding the efficacy of online, noting a scepticism prior to the pandemic.


*‘I was sceptical about the effectiveness at first but after two years I feel it can have the same intensity and effectiveness as face-to-face.’*
(clinician)


*‘I was always opposed to this idea thinking it would really impact the therapeutic relationship. However, I have worked really effectively with many patients and families since with good outcomes with some patients I had never seen in person.’*
(clinician)

Having sessions online allowed clinicians to have insight into young peoples’ home environment and family dynamics, which they would not have seen otherwise. Similarly, parents/caregivers spoke about online therapy providing the clinician with ‘*a real insight into home life*’. Clinicians further reflected on finding new ways of working and engaging with families online. For example, conducting in vivo practice of new behaviours was much easier online than face-to-face as real time support could be provided to families. Being able to access and share relevant documents or visuals to back up a point was also noted as something they would not have at hand in a face-to-face appointment.


*‘I think that I started to use different strategies to engage people on-line—more playful, use of humour—which is easier when families are in their own space and not behaving a certain way because they are in a clinic.’*
(clinician)

All three groups (young people, parents/caregivers, and clinicians) welcomed the opportunity for family members (particularly fathers) to join online sessions who might have not been able to face-to-face. Parents/caregivers could ‘*always be present*’ and were ‘*more able to both attend*’ sessions held online. This brought new opportunities for increased input from different family members, allowing for additional perspectives and systemic work to take place. Parents/caregivers described another advantage of online sessions being the reduced exposure for their child to ‘*other children with severe eating disorder*s’.


*‘I have had more attendance from wider family members than for face-to-face treatment. I think there is also an advantage for some people in that they can more directly translate conversations to home (e.g., meal strategies, self-soothe boxes, etc.)’*
(clinician)


*‘I think the therapist can get a real insight into the patients home life through what goes on around the sessions—ours were chaos with dogs and siblings coming through which often broke the ice when we were at an impasse.’*
(parent/caregiver)


*‘Easier for family therapy as you can organise the whole family to be in one place’*
(young person)

#### 3.4.2. Theme 2: Something Lost

Despite the numerous gains of online therapy, several losses were identified by all three groups. Young people in particular endorsed losses of online therapy. Four subthemes were identified: 2.1. Depth of treatment experience; 2.2. Therapy flow 2.3. Lack of insight into treatment progress; and 2.4 Is this real?

##### Depth of Treatment Experience

Overall, there was a sense that online sessions were perceived as more practical, with a greater focus on things like problem solving and setting tasks. Most described online sessions as being less emotionally focused and less appropriate for more ‘*intense*’ conversations. One clinician wrote that *‘people can avoid difficult emotions/conversations and retreat to their rooms’ (clinician).* Another clinician felt that being online *‘… makes treatments feel more problem-solvey and task focused, rather than emotion focused.’ (clinician).*

Many participants said their ultimate preference for face-to-face sessions was mostly driven by process and relational elements such as rapport building, the feeling of a safer space to express emotions, feeling understood, meaning making, and managing risk and safety for the young person. Several young people noted that the ‘*connection with therapist*’, ‘*body language*’ and the ‘*ability to feel comfortable*’ were unique to face-to-face therapy and were hard to replicate online. There were some young people, however, with alternate views, who found the practical aspects of online sessions helped them to engage with treatment and foster progress.


*‘[I] don’t feel that I can open up as much when just looking at someone one a screen.’*
(young person)


*‘I strongly believe nuances of communication are lost online leading to a lower level of understanding of what is going on for my daughter.’*
(parent/caregiver)


*‘I cannot always easily sense the family’s emotional experience and dynamics during a virtual session especially if a family member is not as engaged.’*
(clinician)

Relatedly, there was also a sense by most that these potential difficulties building trust, connection, and engagement were thought to delay treatment progress and contribute to longer treatment length. Specifically, it was thought that *‘building a therapeutic relationship takes longer [online]’ (parent/caregiver)*.


*‘Face-to-face contribute [s to a] speedier recovery—more connected.’*
(parent/caregiver)


*‘Online therapy may result in slower recovery … less connection.’*
(young person)


*‘…it has felt like, the treatment was longer.’*
(clinician)

##### Therapy Flow

Issues surrounding technology, confidentiality, and the level of engagement of young people seemed to compromise the flow of online sessions, whereas therapy delivered face-to-face ‘*flowed*’ better. At a process level, some parents/caregivers noted that the nuances of non-verbal behaviour (i.e., mannerisms, eye contact, etc.) and communication were lost by the use of technology and that issues relating to the sessions being online (i.e., issues with technology not working) could be very stressful.

For many participants, there were no, or minimal, technical issues, and these were not considered to impact on treatment experience. However, some young people, parents/caregivers, and clinicians noted that technological issues like *‘problems with wifi’, ’[Microsoft] Teams not working’, ‘sound and or picture drops out’,* and ‘*glitches/lost signals delay*’ could significantly disrupt sessions. There were also mentions from parents/caregivers of ‘*technical difficulties*’ such as the ‘*size of our screen’, ‘sound, connectivity*’, and not being able to see each other’s ‘*facial reactions*’ when online. One young person wrote that *‘online therapy may result [in] miscommunication [and a] lack of understanding’ (young person).*

Parents/caregivers and young people further noted difficulties with confidentiality and concerns regarding not having an appropriate therapy space. For example, one parent/caregiver said we were ‘*doing it in our kitchen, [which] was quite confusing because it is not a safe private space*’ with distractions ‘*like the dog, door knocking*’. One parent/caregiver noted that it ‘*can be difficult to have a quick private chat during the session’* when online. Clinicians also suspected confidentiality to be an issue for families where private space is limited.


*‘I sometimes was a bit anxious to tell her [the therapist] how I felt because I was worried someone in my family would be listening as it was online meaning I was in my home.’*
(young person)


*‘Confidentiality is tough—we live in a small flat. My child can’t speak freely without fearing being overheard.’*
(parent/caregiver)


*‘Confidentiality worries due to sharing an environment with others might have an impact on individual sessions if there are restrictions on availability of a separate room’*
(clinician)

Parents/caregivers also voiced their concerns associated with sustaining young peoples’ levels of engagement and participation. Both young people and parents/caregivers noted that young people engaged better with face-to-face sessions. Parents/caregivers noted the challenge associated with ‘*getting my daughter into the room*’ for online therapy and how it was easier for their child ‘*to disengage*’. Young people further commented that with online sessions they did not always feel listened to, could be easily distracted and struggle to build trust with their therapist. Some parents/caregivers also noted that ‘*conversation didn’t flow as easily*’ and that therapy ‘*feels a bit distant*’. Similarly, clinicians also spoke about the challenges of engaging young people online, particularly those with specific, additional needs (e.g., ADHD).


*‘It’s easier for me to talk face-to-face. Online I get distracted and lose interest’*
(young person)


*‘It can be more difficult to fully open up as it’s a less intimate setting. Also, quite easy to get distracted or impatient for the session to end when at home.’*
(young person)


*‘My child tends to look away and shut down online—she does not do this as much in a room and if she does you can bring her around easier’*
(parent/caregiver)

##### Lack of Insight into Progress

Participants shared concerns that proper insight into the young person’s illness severity and treatment progress could have been compromised when sessions were online. In particular, the lack of accurate physical monitoring was described by all participants as a shortfall in online sessions. Furthermore, concealment of illness behaviours were thought to lead to a reduced understanding of a young person’s progress and ultimately resulted in a more prolonged length of treatment. Many parents/caregivers and some young people noted that physical monitoring when sessions are online was not as regular or carried out as accurately at home compared to in the clinic. This impacted people’s perceptions of treatment progress. There was a shared apprehension by parents/caregivers in managing physical health online. All three groups also discussed how clinicians did not have full access to the young person’s appearance, which was problematic when assessing their physical state and detecting any changes, such as weight loss. Furthermore, all three groups discussed how easily people could conceal aspects of the illness, including physical state, when treatment was online.


*‘It was a bit more distressing to be weighed at home than in the clinic’*
(young person)


*‘It was easier for my eating disorder to convince me to lie’*
(young person)


*‘Lack of ability to read body language and gauge mood as accurately. Inability to weigh people, physically examine them, take bloods, do ECGs, etc. leading to lack of accurate knowledge about actual state of someone’s health’*
(parent/caregiver)


*‘It might not be very easy to visually see the changes in patient’s physical state. It might make it easier to falsify weight by patients with eating disorders’*
(clinician)

##### Is this Real? Devaluation vs. Realness

Participants across the three groups questioned the ‘*realness*’ of online sessions. A number of young people and parents/caregivers mentioned how online therapy did not feel ‘*real*’ or personal. Furthermore, young people noted that they were not being taken seriously or could not take online sessions seriously themselves. Similar devaluation comments questioning the ‘*realness*’ of online sessions were made by parents/caregivers, whereas clinicians noted how some families referred to online sessions as ‘*calls*’ or ‘*meetings*’ and were more likely to cancel an online session. Clinicians themselves also struggled to come to terms with the ‘realness’ of the families they met online. Lack of realness also raised questions by some parents/caregivers as to whether online sessions were impactful, and boundaries were easily blurred. Face-to-face human connection was considered irreplaceable by some.


*‘Not feeling my issues were being taken seriously and not taking the advice given as seriously when online’*
(young person)


*‘… doesn’t feel as ‘real’ as face-to-face’*
(parent/caregiver)


*‘One of the strangest things has been saying goodbye to people online. Sometimes it makes it seem like they weren’t even real.’*
(clinician)

#### 3.4.3. Theme 3: One Size Does Not Fit All

All three groups discussed how online therapy would be the best option for some families under certain situations at certain times in treatment. Knowing when online therapy should be offered and to which families at what time in treatment was considered very important. Two subthemes were identified that captured these points: dynamic flexibility (3.1) and timing (3.2)**.**

##### Dynamic Flexibility

All groups discussed how the choice, preference, and acceptability of treatment mode (online vs. face-to-face) were determined by various factors that needed to be considered and reviewed iteratively throughout treatment. These factors included the appropriateness of the specific home setting of each individual family for online sessions, illness severity and associated risk, the need for physical monitoring, and model of treatment to be delivered (e.g., group, individual, and family work).

Clinicians highlighted the need to regularly assess which mode of treatment delivery best suited each individual and family and their unique characteristics. For example, the level of risk and young person’s engagement levels may indicate the need to switch from online to face-to-face, but the reverse may also be considered for another family. As one clinician wrote *‘the riskier the presentation the less safe [I] will feel to review them only online*’.

Taking families into account, accessibility was highlighted as important, as not all families were able to access the internet or had the resources or a suitable/confidential home environment for online therapy. Clinicians also reported that slowed or stalled treatment may prompt the switch from online to face-to-face sessions. Online sessions were considered less robust in these instances. For example, one clinician wrote ‘*If there is failure to respond to treatment it might be important to try a change in format to see whether this will improve response’ (clinician)*. Online therapy was also viewed by clinicians to be less suitable for delivering group and multi-family therapy compared to individual or single-family therapy. Furthermore, some considered online therapy to be preferable for some people. One clinician noted: ‘*patients with ASD [Autism] might feel more confident in their own environment and more forthcoming’ (clinician).*

Many participants noted that young person and/or family’s preference for online or face-to-face therapy is important to consider but should not be the *only* driving force in deciding which mode of treatment delivery is most appropriate. Presentation, risk, level of engagement and treatment progress need to be taken into consideration as well. One clinician said the decision on treatment mode needs to be *‘formulation based—severity of difficulties, neurobiology and temperament, level of parental anxiety’ (clinician).* What the family wants may also be different to their needs. In line with this, some clinicians commented that online therapy is not for every family.


*‘I feel it can be effective for some patients but not all. I am less confident drawing out formulations and other diagrams online’*
(clinician)


*‘Online would not work for my child, she needed 100% face-to-face (I suppose it depends on the child).’*
(parent/caregiver)


*‘I think groups for young people would probably work better in person- it’s easier to be a spectator rather than a participant in an online group. For family and individual work I find both types of treatment fairly equitable’*
(clinician)


*‘I am more likely to offer an individual, skills-based treatment (e.g., CBT) online rather than face-to-face’*
(clinician)

##### Timing

The appropriateness of online or face-to-face therapy was thought by many to be dependent on the stage of treatment. Aside from one young person who found joining the first session (assessment) online easier, face-to-face sessions were deemed necessary by all three groups for the first appointment in order to build trust, assess physical health, and connect with the clinician. Similarly, face-to-face sessions were favoured for the final session. Parents/caregivers and clinicians were more open to the idea of having online sessions at the mid and late stages of treatment on the assumption that there was progress happening and risk was low.


*‘I think it is important to meet your therapist face-to-face occasionally (especially at the beginning of treatment when needing to monitor weight and physical health)…’*
(young person)


*‘…to promote engagement and early behaviour change then face-to-face can be preferable for some young people and families. Towards the end online can work better in terms of returning to normal life and not missing school etc’*
(clinician)


*‘Initial stages of treatment where there could be prominent physical or psychological risk or any stage when risk is increasing, can change the appropriate format for a given patient and requires flexibility.’*
(clinician)

## 4. Discussion

The current study aimed to assess young person, parent/caregiver, and clinician experiences and preferences for online and face-to-face child and adolescent eating disorder treatment. This was considered important given online treatment delivery was enforced during the COVID-19 pandemic, whereas there is more choice in treatment mode now that COVID-19 restrictions on face-to-face working have ended. The current findings indicate that preferences are largely unchanged since the early days of the pandemic in 2020 and that there is still, generally, a preference from young people and parents/caregivers for most of their treatment to be offered face-to-face. Specifically, there was a strong preference for the assessment session, early treatment sessions, and final session to be offered face-to-face. The middle and latter parts of treatment were a time when preferences shifted to a more hybrid mode of delivery, particularly for parents/caregivers.

Online treatment was considered helpful and acceptable, with the overall experience of online treatment rated relatively well. Most participants described experiencing some benefits from online therapy and were able to work with this mode of delivery. However, when directly compared to face-to-face treatment, online was typically rated more poorly, or equivalent at best. Specifically, young people and parents/caregivers both reported finding engagement with their therapist and the ability to talk about emotional difficulties easier during face-to-face sessions.

When comparing the online treatment experience in 2020 to 2022, there were relatively few differences. On all domains assessed, there were no significant differences in the young person’s experience of online treatment. The largest change for parents/caregivers and clinicians, but not young people, was the impact of technology on the treatment experience, with parents/caregivers reporting a significant reduction in the impact of technology on their online treatment experiences. Furthermore, there was a slight, but significant, increase in how well parents/caregivers perceived online treatment to be able to address the important issues in therapy. Relatedly, clinicians said they felt significantly more comfortable during online treatment. Apart from these few domains, there were no other significant differences in the experience of online therapy from 2020 to 2022.

While it could be hypothesised that there would be an overall shift towards favouring online therapy due to the increased exposure and practice of being online in most life domains (social, education, employment, etc.), this is not supported by the current data. What the data does suggest, however, is that people are willing to *accept* online treatment, even if they do not necessarily *prefer* it. This is particularly evident in the themes described from the reflexive thematic analysis of free-text responses. The quotes, themes, and subthemes described in this study are also strikingly similar to those reported in the initial study [16].

What is apparent from the current data is the importance for people and services to move beyond the binary of online *or* face-to-face and consider how both (and potentially other formats) could be used flexibly for the same person/family based on presentation, resources, preference, and treatment stage. It is now a widespread practice in the UK for both modalities to be offered. The current data provides valuable insights into *when, why*, and *how* best to choose which modality to use. Especially given data suggest outcomes could be equivalent in both formats [11].

For practitioners, the current findings suggest face-to-face sessions are preferred at the beginning, early stages, and final session of treatment, if available. This could then be transitioned to a mix of online and face-to-face during the middle and latter parts of treatment. Indicators for either persisting with face-to-face treatment and/or returning to face-to-face if online treatment sessions have begun, might be limited or poor individual/family engagement, minimal treatment progress, and/or mental or physical health deterioration. Conversely, online sessions might be indicated as an alternative to face-to-face sessions during treatment if in vivo practice of new learning (e.g., managing meals and/or distress) is needed or if face-to-face treatment access is very difficult.

Looking ahead, it will be important to try and determine whether people’s perceptions, experiences, and preferences for treatment mode have a direct impact on treatment engagement, outcomes, and duration. Available data suggests the therapeutic alliance and early symptoms change both significantly impact upon end of treatment outcomes [38,39]. Given current data that online treatment, especially if provided in the early stages of treatment, is perceived to delay engagement and lengthen treatment, it will be important to quantitatively determine whether this translates into poorer outcomes. Another major consideration for future research will be to try and understand the limits of online working, especially given the high medical comorbidity and need for physical health examinations as part of treatment.

### Strengths and Limitations

The mixed method design and inclusion of the different perspectives of young people, parent/caregivers, and clinicians created a richness to the data. This helped to understand and compare the experiences and preferences of those involved in treatment. Additionally, anonymity during data collection potentially allowed for greater honesty in people’s responses and buffered against any power dynamics in eliciting feedback from service users and staff.

There are, however, several important limitations to the current study. The sample size remains relatively small, particularly for young people, making any conclusions drawn from the current study tentative. It will be important to see if these data are replicated in other services, settings (e.g., day programme treatment), and other countries. The anonymous design also meant demographic information, such as age, sex, gender, and other potentially important information that could influence the relationship of the young person, parent/caregiver, and/or clinician with online treatment, could not be reported.

While it is a strength of the current data to be able to compare current and past experiences within the same service, it also limits the generalisability of the current data. More data are needed from other services to determine whether these experiences and preferences are shared or unique to one clinic. An additional limitation was that email addresses were not available for every eligible young person and parent/caregiver. Recruitment partly relied on one family member forwarding the survey link onto other eligible participants within their family. As such, selection bias may be present and eligible participants’ experiences not represented in the current findings.

Additionally, none of the current preference/experience data can be linked to outcome data. It will be important for future studies to focus on whether online, face-to-face, or hybrid treatment impacts outcome. Lastly, some of the young people and parents/caregivers were relatively early in their treatment journey (<5 sessions attended), meaning the majority of their data were regarding their preference, rather than experiences. These are two distinct factors and more specific data on each would be useful to collect.

## 5. Conclusions

Current findings suggest that young person, parent/caregiver, and clinicians’ experiences and preferences for online working were largely unchanged since the beginning of the COVID-19 pandemic, when treatments were shifted abruptly online. Online treatment continues to be considered helpful and acceptable. However, there remains a strong preference for the assessment session, early treatment sessions, and final session to be offered face-to-face. The middle and latter parts of treatment is a time when preferences seem to shift slightly to a more hybrid mode of delivery, particularly for parents/caregivers. Participants in this study reported finding engagement with the therapist and the ability to talk about emotional difficulties easier during face-to-face treatment. Stepping away from the binary of online *or* face-to-ace, the current data suggest that a hybrid and flexible model is a way forward, and this study provides insights into how this might be structured.

## Figures and Tables

**Figure 1 nutrients-15-03732-f001:**
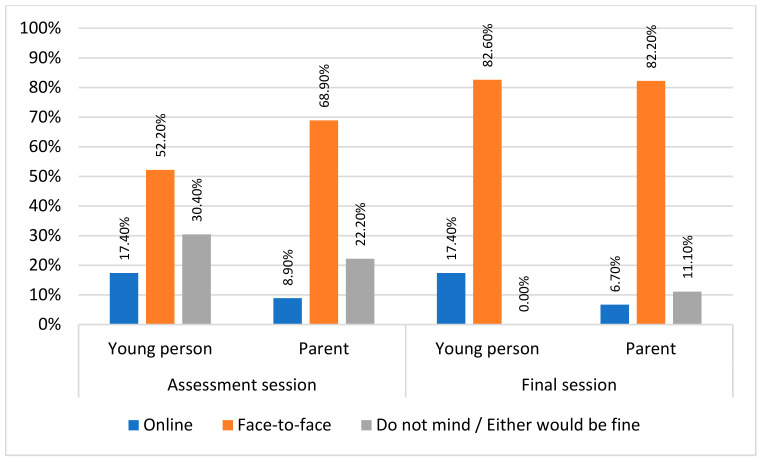
Young person and parents’/caregivers’ preferences for treatment mode for the assessment and ending sessions.

**Figure 2 nutrients-15-03732-f002:**
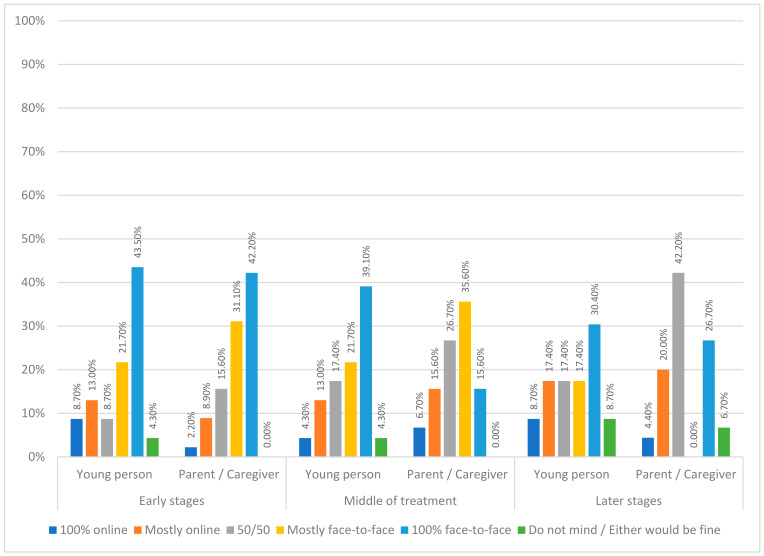
Young person and parent/caregiver preferences for treatment mode at the early, middle, and late stages of treatment.

**Table 1 nutrients-15-03732-t001:** Self-reported proportion of treatment received online versus face-to-face.

Treatment Mode	Young People % (n)	Parents/Caregivers % (n)
100% online	16.00% (4)	12.20% (6)
Mostly online	28.00% (7)	26.50% (13)
50/50	8.00% (2)	12.20% (6)
Mostly face-to-face	36.00% (9)	40.80% (20)
100% face-to-face	12.00% (3)	8.20% (4)
*Missing*	0	0

**Table 2 nutrients-15-03732-t002:** Young person and parents’/caregivers’ experiences of online treatment in 2022.

	Young People (N = 25)		Parents/Caregivers (N = 49)		Test Statistics
	Mdn. (IQR)	n	Mdn. (IQR)	n	
Overall experience	5 (4–6)	22	6 (5–7)	45	*U* = 330.00, *p* = 0.24
Difficulties understood by therapist	6 (5–6)	23	7 (6–7)	45	*U* = 295.50, *p* = 0.002 *
Address important issues	6 (5–7)	23	7 (6–7)	45	*U* = 394.00, *p* = 0.83
Impact of technology on treatment experience	2 (1–4)	18	3.5 (1–4)	42	*U* = 334.50, *p* = 0.47
Benefit from online therapy	6 (4–7)	15	6 (5–7)	36	*U* = 262.00, *p* = 0.86
Overall experience	5 (4–6)	22	6 (5–7)	45	U = 330.00, *p* = 0.24

Note: All ratings used scale 1–7 (1 = lowest possible negative score; and 7 = highest possible positive score). * = *p* < 0.05

**Table 3 nutrients-15-03732-t003:** Overall preferences for treatment mode.

	Young People % (n)	Parents/Caregivers % (n)	Clinicians % (n)
100% online	4.30% (1)	0.00% (0)	0.00% (0)
Mostly online	13.00% (3)	15.60% (7)	6.30% (1)
50/50	21.70% (5)	24.40% (11)	37.50% (6)
Mostly face-to-face	26.10% (6)	35.60% (16)	31.30% (5)
100% face-to-face	34.80% (8)	24.40% (11)	0.00% (0)
*Missing*	2	4	4

**Table 4 nutrients-15-03732-t004:** Young person and parent/caregiver perception of the impact of treatment mode on different aspects of treatment.

	Online Better % (n)		Both Equally Good % (n)		Face-to-Face Better % (n)	
	Young Person	Parent/ Caregiver	Young Person	Parent/ Caregiver	Young Person	Parent/ Caregiver
Engagement with therapist	0.00% (0)	5.00% (2)	29.40% (5)	42.5% (17)	70.60% (12)	52.50% (21)
Confidentiality	17.60% (3)	5.00% (2)	58.80% (10)	67.5% (27)	23.50% (4)	27.50% (11)
Ability to discuss emotional difficulties	11.80% (2)	5.00% (2)	41.20% (7)	35.0% (14)	47.10% (8)	60.00% (24)
Ability to discuss practical difficulties	17.60% (3)	5.00% (2)	47.10% (8)	35.0% (14)	35.30% (6)	60.00% (24)
Speed of recovery	11.80% (2)	5.00% (2)	47.10% (8)	52.50% (21)	41.20% (7)	42.50% (17)
Involvement of family	25.00% (4)	17.50% (7)	43.80% (7)	50.00% (20)	31.30% (5)	32.50% (13)

**Table 5 nutrients-15-03732-t005:** Comparison between 2020 and 2022 cohorts regarding online therapy experience.

	2020 Sample ^		2022 Sample		Test Statistic
	Mdn. (IQR)	N	Mdn. (IQR)	N	
**Clinicians**		**N = 23**		**N = 16**	
Self-efficacy	7 (6–8)	23	8 (7.25–8)	16	*U* = 130.50, *p* = 0.13
Self-efficacy compared to face-to-face	4 (3–6)	23	4.5 (3–6)	16	*U* = 175.00, *p* = 0.81
Level of comfort	4 (3–6)	23	9 (8.25–10)	16	*U* = 18.00, *p* < 0.001 **
Comfort compared to face-to-face	4 (3–6)	23	5 (5–7)	16	*U* = 119.50, *p* = 0.07
Impact of technology on treatment experience	4 (2–6)	23	5 (2–7.75)	16	*U* = 174.00, *p* = 0.79
**Young People**		**N = 53**		**N = 25**	
Overall experience	5 (4–6)	53	5 (4–6)	22	*U* = 491.00, *p* = 0.27
Difficulties understood by therapist	6 (4.25–7)	52	6 (5–6)	23	*U* = 572.50, *p* = 0.76
Address important issues	5 (4–6)	52	6 (5–7)	23	*U* = 446.50, *p* = 0.07
Impact of technology on treatment experience	3 (2–4)	53	2 (1–4)	18	*U* = 348.00, *p* = 0.08
Benefit from online therapy	5 (3.75–7)	46	6 (4–7)	15	*U* = 305.50, *p* = 0.50
**Parents/Caregivers**		**N = 75**		**N = 49**	
Overall experience	6 (4.75–7)	70	6 (5–7)	45	*U* = 1565.00, *p* = 0.95
Difficulties understood by therapist	7 (6–7)	70	7 (6–7)	45	*U* = 1501.00, *p* = 0.64
Address important issues	6 (5–7)	71	7 (6–7)	45	*U* = 1160.50, *p* = 0.01 *
Impact of technology on treatment experience	7 (5–7)	58	3.5 (1–4)	42	*U* = 420.50, *p* < 0.001 **
Benefit from online therapy	7 (5–7)	58	6 (5–7)	36	*U* = 951.50, *p* = 0.45

Note: Ratings for young people and parent/caregiver surveys used a 1–7 scale (1 = lowest possible negative score; and 7 = highest possible positive score). Ratings for clinicians used a 1–10 scale (1 = lowest possible negative score; and 10 = highest possible positive score). * = *p* < 0.05; ** = *p* < 0.001; ^ Data originally reported by Stewart et al. [16].

## Data Availability

Data are available from the corresponding author upon reasonable request.

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
