# Peer review of "Do People Want the ‘New Normal’? A Mixed Method Investigation of Young Person, Parent, and Clinician Experience and Preferences for Eating Disorder Treatment Delivery in the Post-COVID-19 World"

_nutrients, 2023, doi:10.3390/nu15173732_

Round 1

Reviewer 1 Report

. On the whole, I found the article to be well-crafted and insightful. However, I do have a few comments and suggestions for enhancement.

Introduction:

1. I would appreciate it if you could provide an explanation regarding the novelty of your research. Additionally, I am curious to know if similar studies have been conducted previously, and if you are confident that your research introduces new and original findings.

Methods:

1. It would be appropriate to include information about the research design in the "Methods" section.

2. Could you kindly confirm whether the work experience of the clinic staff was considered in the inclusion criteria?

3. Considering that the prior study was conducted in the Maudsley Centre for Child and Adolescent Eating Disorders, it might have been worth considering the possibility of broadening the geographical coverage for this research.

4. May I inquire about the number of potential respondents to whom the link was sent? Additionally, could you please provide the response rate upon receiving answers to the online survey?

5. I was wondering if the calculation of the required sample size was performed for the study. If it was, could you please provide more information on the determined sample size needed to obtain representative data?

Results:

6. Regarding the table 2 "Overall experience" variable. While it is recognized as a Likert scale, could you please specify what aspect it evaluates, such as satisfaction, time in days, or any other dimension?

7. Considering the data presented in Table 3, could it be suggested that the number of young respondents may be limited to draw definitive conclusions about patient preferences?

8. In order to improve the data visualization, I recommend considering some format changes for figures 2 and 3. Doing so could enhance the overall quality of your article. The colors used in the legends for the tables are somewhat difficult to interpret. Furthermore, I would appreciate some clarification regarding the y-axis in figure 2, as it seems to have a maximum value of 50%.

9. I would appreciate it if you could provide information on the software tools employed for the statistical analysis. Moreover, could you kindly provid the program you used to create the figures?

10. I would like to inquire if you utilized specialized software for text analysis in the qualitative research section. Such software is known to facilitate text search, coding, and analysis.

Author Response

Response to reviewer 1

On the whole, I found the article to be well-crafted and insightful. However, I do have a few comments and suggestions for enhancement.

Thank you for your kind words. We have attempted to address your comments comprehensively below.

Introduction:

  1. I would appreciate it if you could provide an explanation regarding the novelty of your research. Additionally, I am curious to know if similar studies have been conducted previously, and if you are confident that your research introduces new and original findings.

We have attempted to address this in the two final paragraphs of the introduction. To emphasise the need for and novelty of the current study we have added the following:

The authors are not aware of any published studies reporting the experience of online working in child and adolescent services post the easing of COVID-19 restrictions. (line 82)

Methods:

  1. It would be appropriate to include information about the research design in the "Methods" section.

Thank you for your comment. The following section has now been added to the Materials and Methods section

Study design

This cross-sectional study employed an online survey methodology to anonymously gather information about the experience of and preferences for online and face-to-face treatment. The use of online surveys to collect both quantitative and qualitative data was considered most appropriate given anonymity of participants was preserved and disclosure of ones’ own preferences encouraged … (line 98)

  1. Could you kindly confirm whether the work experience of the clinic staff was considered in the inclusion criteria?

Apologies this was unclear. The only eligibility criteria for the study were that the clinicians were a) working at MCCAED and b) were a registered health professional. To clarify this we have amended the text as follows:

All clinicians who were a) employed at MCCAED at the time of data collection and b) were a registered mental health professional were eligible for this study and invited to participate. Clinicians were all fully qualified, registered mental health professionals including clinical psychologists, family therapists, medical doctors and psychiatrists all of whom had experience in the treatment of eating disorders. Level of experience was not included as an eligibility criterion to participate, rather, these data were collected and reported on. (line 114)

  1. Considering that the prior study was conducted in the Maudsley Centre for Child and Adolescent Eating Disorders, it might have been worth considering the possibility of broadening the geographical coverage for this research.

We agree this would be useful and interesting data to collect. We have added this as a limitation of the current study. See amended text:

While it is a strength of the current data to be able to compare current and past experiences within the same service, it limits the generalisability of the current data. More data is needed from other services to determine whether these experiences and preferences are shared or unique to one clinic. (line 688)

  1. May I inquire about the number of potential respondents to whom the link was sent? Additionally, could you please provide the response rate upon receiving answers to the online survey?

Survey links were sent to clinicians and the main contact for each family, which was predominantly mothers. Given the anonymous design, it is not possible to determine how many young people, mothers, fathers, other types of caregivers were specifically invited, nor whether the main contact who was written to extend the invitation to some or all family members who participated in treatment.

We have now amended the text to clarify this to the reader. See text:

The available email addresses on file for eligible family members were used for recruitment purposes. This was often a single email address per family for a parent/caregiver. As such, the email invitation requested the survey be sent to their young person and/or partner (where applicable) if they met the inclusion criteria and the deemed it appropriate.  (line 142)

We have also added this as a limitation to the current study:

An additional limitation was that email addresses were not available for every eligible young person and parent/caregiver. Recruitment partly relied on one family member forwarding the survey link onto eligible participants. As such, selection bias may be present and eligible participants’ experiences not represented in the current findings. (line 690)

  1. I was wondering if the calculation of the required sample size was performed for the study. If it was, could you please provide more information on the determined sample size needed to obtain representative data?

Unfortunately, a sample size calculation was not conducted. Rather, we attempted to contact all eligible participants possible. We have emphasised in the study limitations the following:

The sample size remains relatively small, particularly for young people, making any conclusions drawn from the current study tentative. (line 681)

Results:

  1. Regarding the table 2 "Overall experience" variable. While it is recognized as a Likert scale, could you please specify what aspect it evaluates, such as satisfaction, time in days, or any other dimension?

Thank you for raising this. This item was intended to ask people to give a rating of their experience of online treatment ‘on the whole’. We did not specify what this should include, rather we wanted the person’s general impression. To clarify this to the reader we have amended the text as follows:

Median scores for the overall experience (the subjective general impression of online treatment), the ability to be understood by the therapist and address important issue, and the overall benefit were all rated (line 216)

  1. Considering the data presented in Table 3, could it be suggested that the number of young respondents may be limited to draw definitive conclusions about patient preferences?

We completely agree. As per point 5 above we have added the following to the limitations section:

The sample size remains relatively small, particularly for young people, making any conclusions drawn from the current study tentative. (line 681)

  1. In order to improve the data visualization, I recommend considering some format changes for figures 2 and 3. Doing so could enhance the overall quality of your article. The colors used in the legends for the tables are somewhat difficult to interpret. Furthermore, I would appreciate some clarification regarding the y-axis in figure 2, as it seems to have a maximum value of 50%.

Apologies this was unclear. We have now made the figures colour to enhance readability. We have also extended the y-axis of Figure 2 to go from 0 – 100%. Lastly, we have added data labels for ease of interpretation.

  1. I would appreciate it if you could provide information on the software tools employed for the statistical analysis. Moreover, could you kindly provide the program you used to create the figures?

Thank you for highlighting that this was missing. We have now added the following:

Quantitative data were analysed using Statistical Package for the Social Sciences (SPSS Version 29 (IBM, 2019). No software was used during the reflexive thematic analysis. Figures were created using Microsoft Excel (Microsoft Corporation, 2018). (line 166)

  1. I would like to inquire if you utilized specialized software for text analysis in the qualitative research section. Such software is known to facilitate text search, coding, and analysis.

Again, apologies this was missing in the original manuscript. As per point 9 above, we did not use any software during the qualitative reflexive thematic analysis and have specified this in the manuscript now.

Reviewer 2 Report

General comments 

- Authors need to standardize the system of calling references. Sometimes they use numbers, and sometimes "author, date". 

Specific comments

Abstract 

- Authors could include numerical values for quantitative results.

- Keywords have numbers wrongly included. 

Background 

- The introduction is fantastic and makes it clear what the gap in scientific knowledge is. However, the authors end the introduction with a question ("Now that people don't have to do their treatment online, do they like it?"). I suggest that the authors transform this sentence into a clearer objective and that it is in interrogative form. 

Materials and methods 

- The authors do not report how they obtained consent to include participants, especially as participants were under 18 years old. This should be set out in this 'Ethical aspects' section. 

- The authors should include information on which variables were collected and taken into account for this study. There needs to be more than citing the previous work. The current text could be clearer and compromise replication. 

- I suggest including a topic "Study design", in which the study information should be clear. In this case, a longitudinal, qualitative, and quantitative study, among other details. 

Results 

- Authors should standardize the way of reporting decimal numbers, 2 or 3 numbers after the point.

Discussion 

- In the study's limitations, the authors should report the absence of demographic data for this sample. Age, sex, gender, and other issues could be considered variables that could influence the relationship of the patient, parents, and clinician with the online treatment.

Author Response

Response to reviewer 2

Authors need to standardize the system of calling references. Sometimes they use numbers, and sometimes "author, date". 

 Apologies for this oversight. This has now been amended and is consistent throughout the manuscript.

Specific comments

Abstract 

  1. Authors could include numerical values for quantitative results.

Thank you for this suggestion. We have now added numerical numbers to the abstract. It now reads:

Abstract: Eating disorder treatment was predominantly provided online during the COVID-19 pandemic, which has continued into the post-pandemic world. This mixed method study explored young person, parent/caregiver and clinician experiences of child and adolescent eating disorder treatment. In total, 90 participants (25 young people, 49 parents/caregivers, 16 clinicians) completed online surveys about the experience of online working. Data were compared to similar data collected by the same service earlier in the pandemic. Result show preferences are largely unchanged since 2020. Online treatment is considered helpful and acceptable by all groups. Nevertheless, face-to-face assessment sessions (young people: 52.2%; parents: 68.9%) and final sessions (young people: 82.6%; parents: 82.2%) were preferred compared to online. There was also a preference for early treatment sessions to either be always or mostly face-to-face (young people: 65.2%; parents: 73.3%). The middle and latter parts of treatment were a time when preferences shifted slightly to a more hybrid mode of delivery. Participants reported finding engagement with the therapist (young people: 70.6%; parents: 52.5%) easier during face-to-face treatment. Stepping away from the binary of online or face-to-face, the current data suggests a hybrid and flexible model is a way forward and provides insights into how to structure this.  (line 13)

  1. Keywords have numbers wrongly included. 

Thank you for pointing this out. Numbers have now been deleted.

Background 

  1. The introduction is fantastic and makes it clear what the gap in scientific knowledge is. However, the authors end the introduction with a question ("Now that people don't have to do their treatment online, do they like it?"). I suggest that the authors transform this sentence into a clearer objective and that it is in interrogative form. 

Apologies this was unclear and too conversational. We have now amended the aims of the study, which now read:

The primary aim of this study was to assess young person, parent/caregiver and clinician current experiences of online specialist child and adolescent eating disorder treatment now that COVID-19 restrictions on face-to-face meetings have eased. The secondary aim was to compare this to similar data previously collected by the same service in the early parts of the COVID-19 pandemic. (line 90)

Materials and methods 

  1. The authors do not report how they obtained consent to include participants, especially as participants were under 18 years old. This should be set out in this 'Ethical aspects' section. 

This study was approved as a clinical audit and service evaluation project. Clinical audits and service evaluation projects are approved by an institutional review board review that involves peer review and committee approval. This approval covers data collection, analysis and publication of findings without formal ethical approval or consent from participants. The original study (Stewart, et al., 2021) was published under this type of approval, as have other recent studies in Nutrients. For example, Billich, et al. 2022 (https://doi.org/10.3390/nu14163304); Visser, et al., 2022 (https://doi.org/10.3390/nu15010154).

Although not explicitly required by clinical audit and service evaluation approval, to ensure this study was conducted ethically, the authors informed participants of the study purpose and design and all were provided opportunity to speak to the project team with any queries.

To clarify this the text has been updated as follows:

This study was approved by the South London and Maudsley (SLaM) child and adolescent mental health services (CAMHS) Service Evaluation and Audit Committee (Project #: 2022-02). As this study constitutes clinical audit or service evaluation, NHS Research Ethics Committee approval was not required. SLaM CAMHS service evaluation and audit approval allows for analysis and publication of anonymised data collected as part of approved projects without written consent from participants or caregivers. Nevertheless, all participants were informed of the design and purpose of the study and provided opportunity to speak to project team with any queries. No young person was invited to participate without parental awareness of the study. It was made explicit to all that participation was voluntary. All methods were performed in accordance with the stipulated guidelines and regulations. (Line 127)

  1. The authors should include information on which variables were collected and taken into account for this study. There needs to be more than citing the previous work. The current text could be clearer and compromise replication. 

Thank you for raising this suggestion. To ensure transparency and replicability we have added the survey questionnaires as supplementary material.

  1. I suggest including a topic "Study design", in which the study information should be clear. In this case, a longitudinal, qualitative, and quantitative study, among other details. 

 Thank you for this suggestion. We have now added a ‘study design’ section. See line 92

The procedure section has also been updated to give more detail on the recruitment process. See line 142 of revised manuscript)

Results 

  1. Authors should standardize the way of reporting decimal numbers, 2 or 3 numbers after the point.

Thank you for raising this inconsistency. We have now amended the manuscript so that numbers have 2 decimal places. The only exception is Likhert scale items that have no decimals as they represent an exact number.

 Discussion 

  1. In the study's limitations, the authors should report the absence of demographic data for this sample. Age, sex, gender, and other issues could be considered variables that could influence the relationship of the patient, parents, and clinician with the online treatment.

Thank you for this suggestion. This has now been added to the limitations section.

The anonymous design also meant demographic information, such as age, sex, gender, and other potentially important that could influence the relationship of the young person, parent/caregiver and/or client with online treatment cannot be reported. (line 684)

Reviewer 3 Report

The study is interesting but I have a few comments

1. there are numbers next to the keywords - for what purpose? The Nutrients template does not require this. 

2) The purpose of the study is convoluted and may not be understood by the reader. 

3. what is the endpoint of the study? 

4. the inclusion criteria should be clearly stated

5. the authors did not provide the bioethics committee approval number

6. will the authors give practical implications as a result of the study? 

The paper in general is interesting but I don't know what its purpose is. Treatment of eating disorders cannot always be done online. Which disease entities did the authors have in mind? What about patients who were in a health or even life-threatening condition as a result of anorexia? The paper needs to be detailed. 

Author Response

Response to reviewer 3

The study is interesting, but I have a few comments

  1. there are numbers next to the keywords - for what purpose? The Nutrients template does not

Thank you for pointing this out. Numbers have now been deleted.

  1. The purpose of the study is convoluted and may not be understood by the reader. 

Apologies this was unclear. As per response to previous reviewer comments we have now changed the final sentence of the introduction to state primary and secondary aims of the study. They read:

The primary aim of this study was to assess young person, parent/caregiver and clinician current experiences of online specialist child and adolescent eating disorder treatment now that COVID-19 restrictions on face-to-face meetings have eased. The secondary aim was to compare this to similar data previously collected by the same service in the early parts of the COVID-19 pandemic. (line 90)

  1. what is the endpoint of the study? 

Apologies this was not in the original manuscript. This has now been added:

Initial email invitations were sent in July 2022 followed by a maximum of three follow-up emails. Recruitment to this study closed in November 2022. (line 145)

  1. the inclusion criteria should be clearly stated

Thank you for highlighting that this was unclear in the original submission. The text has been updated to read:

Inclusion criteria were a) being a young person with a diagnosis of a DSM-5 eating disorder or their parent, b) a current patient of the outpatient service at MCCAED and c) having been in treatment for a minimum of two months. (line 109)

All clinicians who were a) employed at MCCAED at the time of data collection and b) were a registered mental health professional were eligible for this study and invited to participate. (line 114)

  1. the authors did not provide the bioethics committee approval number

Apologies this was missing in the original submission. This has now been provided. See text line 127.

  1. will the authors give practical implications as a result of the study? 

Thank you for this suggestion. It adds something important to the manuscript. We have now added a paragraph of practitioner implications to the discussion. See text:

For practitioners, the current findings suggest face-to-face sessions are to be offered at the beginning, early stages and final session of treatment, if available. This could then be transition to a mix of online and face-to-face during the middle and latter parts of treatment. Indicators for either persisting with face-to-face treatment and/or returning to face-to-face if online treatment sessions have begun, might be limited or poor individual/family engagement, minimal treatment progress, and/or mental or physical health deterioration. Conversely, online sessions might be indicated as an alternative to face-to-face sessions during treatment if in-vivo practice of new learning (e.g. managing meals and/or distress) is needed or if face-to-face treatment access is very difficult.  (line 653)

  1. The paper in general is interesting but I don't know what its purpose is. Treatment of eating disorders cannot always be done online. Which disease entities did the authors have in mind? What about patients who were in a health or even life-threatening condition as a result of anorexia? The paper needs to be detailed. 

We are glad you found the paper interesting. In our experience several eating disorder treatment centres have now moved online – even for underweight anorexia nervosa. The purpose is to understand how families and clinicians experience this shift. We are hoping the amendments to the study aims have clarified this. We have added a few sentences to the introduction to reflect this broader cultural shift towards online working the impact people are reporting more generally:

This seems to match a broader cultural shift internationally for more of our daily interactions to be online. There has been huge increases in the use of online across social, education, vocational and healthcare settings. While beneficial and more convenient in many ways, many have reported a sense of fatigue, exhaustion and disconnection as a result of being online more (Li & Yee, 2023; Nesher Shoshan & Wehrt, 2022). As such it is important to understand people’s experiences of and preferences for being online.  (line 68)

Additionally, to highlight the need for medical examination and the difficulties of this with online working need we have added the following to the discussion:

Another major consideration for future research will be to try and understand the limits of online working, especially given the high medical comorbidity and need for physical health examinations as part of treatment. (line 667)

Round 2

Reviewer 1 Report

All of my comments are considered and there is no more ones.

Reviewer 3 Report

Thank you for your work

minior